# Colitis as the Main Presentation of COVID-19: A Case Report

**DOI:** 10.3390/medicina59030576

**Published:** 2023-03-15

**Authors:** Vlasta Oršić Frič, Vladimir Borzan, Andrej Borzan, Izabela Kiš, Branko Dmitrović, Ivana Roksandić-Križan

**Affiliations:** 1Faculty of Medicine, Josip Juraj Strossmayer University of Osijek, 31000 Osijek, Croatia; 2Department of Gastroenterology and Hepatology, University Hospital Center Osijek, 31000 Osijek, Croatia; 3Department of Surgery, University Hospital Center Osijek, 31000 Osijek, Croatia; 4Department of Pathology and Forensic Medicine, University Hospital Center Osijek, 31000 Osijek, Croatia; 5Department of Clinical Microbiology and Hospital Infections, University Hospital Center Osijek, 31000 Osijek, Croatia

**Keywords:** COVID-19, digestive system infection, gastrointestinal involvement

## Abstract

The main symptoms of coronavirus disease (COVID-19) are fever, cough, tiredness, and loss of smell and taste. Gastrointestinal symptoms are less common. A 38-year-old female patient, previously healthy, presented with a history of hematochezia up to 8 times per day, followed by abdominal cramps, urgency, and chills for two days. She did not have any respiratory symptoms and was previously vaccinated for COVID-19. She was afebrile, with normal vital signs. Blood samples showed normal complete blood count and increased C-reactive protein (CRP), fibrinogen, and D-dimer levels (66 mg/L, 4.1 g/L, and 2302 μ/L FEU, respectively). Stool samples for stool culture, *C. difficile*, and viral examination came back negative. On day 3, she reported a mild cough, fever and loss of smell and taste. Nasopharyngeal swab for SARS-CoV-2 (severe acute respiratory syndrome coronavirus 2) PCR test came back positive. On day 6, the patient still had hematochezia accompanied by abdominal cramps, but fever and respiratory symptoms withdrew. CRP, fibrinogen, and D-dimers were still elevated, as well as liver enzyme levels. Sigmoidoscopy was performed with biopsies taken from sigmoid and rectum for histology and PCR SARS-CoV-2 testing. CT angiography showed no signs of thrombosis in mesenteric veins or arteries. PCR test for SARS-CoV-2 virus from rectal biopsy sample was positive. Patient was treated with methylprednisolone iv for two days and peroral prednisone afterwards, with mesalamine, metronidazole and enoxaparin. Sigmoidoscopy was repeated after two weeks showing only mild hyperemia. At that time, the patient had normal stool, normal CRP, liver enzyme, fibrinogen, and D-dimer levels, and normocytic anemia (hemoglobin level of 103 g/L). We wanted to show that severe gastrointestinal symptoms, such as hemorrhagic colitis, can be the main presentation of COVID-19, even in young patients with no prior comorbidities. In such a case, PCR test in biopsy samples can be performed to prove SARS-CoV-2 infection of bowel mucosa.

## 1. Introduction

In December 2019, novel severe acute respiratory syndrome coronavirus 2 (SARS-CoV-2), which causes coronavirus disease (COVID-19), was identified. The virus rapidly spread around the world causing a global pandemic [1]. The main symptoms of COVID-19 are fever, cough, tiredness, and loss of smell and taste [2]. Gastrointestinal symptoms, such as vomiting and diarrhea, are less common. Here, we present a case of a patient with severe gastrointestinal symptoms as the main clinical presentation of COVID-19.

## 2. Case Presentation

A 38-year-old female patient, previously healthy, presented with a history of hematochezia up to 8 times per day, followed by abdominal cramps, urgency, and chills for two days. She denied having traveled or having ill contacts, and she did not recall taking any food or medication that could be the cause of her symptoms. She did not have any respiratory symptoms and was previously vaccinated for COVID-19. During physical examination, she was afebrile, with a blood pressure of 110/70 mm Hg, pulse of 60 bpm, and oxygen saturation of 98%, and an abdominal examination showed tenderness in the left lower quadrant of the abdomen. Blood samples showed a normal complete blood count, C-reactive protein (CRP) level of 66 mg/L, fibrinogen level of 4.1 g/L, and D-dimer level of 2302 mcg/L FEU. Stool samples for stool culture, *Clostridioides difficile*, and viral examination were taken, and they came back negative. She received azithromycin 500 mg/day for 3 days and rehydration. On day 3 after the first visit, her CRP levels rose to 89 mg/L, and she reported a mild cough, fever, and loss of smell and taste. A nasopharyngeal swab for SARS-CoV-2 polymerase chain reaction (PCR) test came back positive. On day 6, the patient still had hematochezia accompanied by abdominal cramps, but fever and respiratory symptoms withdrew. CRP level, fibrinogen, and D-dimers were still elevated. Additionally, aspartate aminotransferase (AST) and alanine aminotransferase (ALT) became elevated (78 U/L and 61 U/L, respectively). Gastroenterologist was consulted, and sigmoidoscopy was performed. Sigmoid mucosa showed hyperemia, submucosal bleeding, and erosions (Figure 1); changes were less prominent in rectal mucosa. Biopsies were taken from sigmoid colon and rectum for histology and for PCR SARS-CoV-2 testing (FTD SARS-CoV-2 Assay, Fast Track Diagnostics, Luxembourg). Computed tomography (CT) angiography of mesenteric blood vessels was performed and showed no signs of thrombosis in superior or inferior mesenteric vein or artery, and there was no bowel wall thickening. Histology showed plasma cell and neutrophil infiltration in lamina propria, cryptal abscesses, and cryptal destruction (Figure 2). A PCR test for SARS-CoV-2 virus from colonic biopsy sample was positive. The viral load of the respiratory specimen was higher than that of the rectal specimen, based on the Ct values. The patient was treated with methylprednisolone iv. for two days and peroral prednisone afterwards, with mesalamine orally and rectally, metronidazole orally and enoxaparin. On day 11, the patient was feeling better, with 5–6 normal stools per day with traces of blood; the blood samples showed normal CRP, AST and ALT, fibrinogen 3.7 g/L, and D-dimers 1332 mcg/L FEU. A control sigmoidoscopy was performed two weeks after the first one and showed only mucosal hyperemia (Figure 3). The patient eventually recovered completely with no residual symptoms.

## 3. Discussion

Our patient presented with a hematochezia, abdominal pain, and chills. As she was previously vaccinated for COVID-19, and she did not have any respiratory symptoms, COVID-19 infection was not suspected at first as the cause of her symptoms. Firstly, infectious colitis due to other possible pathogens was excluded, including bacterial (*Salmonella* spp., *Shigella* spp., *Escherichia coli*, *Yersinia enterocolitica*, *Campylobacter jejuni*), viral (Norovirus, Adenovirus) and *C. difficile* infection (glutamate dehydrogenase antigen toxin screen). It is now well known that SARS-CoV-2 infection can cause gastrointestinal symptoms in 12% up to more than 60% of patients [3,4], and, in some, they can even precede or be present without respiratory symptoms [5]. Most common gastrointestinal symptoms of COVID-19 are diarrhea, nausea, and vomiting [6], and symptoms of hemorrhagic colitis were reported only in several cases so far [7,8].

SARS-CoV-2 binds to angiotensin-converting enzyme (ACE) 2, which is expressed in the gastrointestinal epithelial cells, among other tissues. Viral nucleocapside protein and viral RNA were also detected in gastrointestinal epithelial cells, proving that SARS-CoV-2 could cause gastrointestinal infection [9]. Qian et al. [10] found coronavirus virions in rectal tissue observed under the electron microscopy, also proving that SARS-CoV-2 can cause active gastrointestinal infection. Furthermore, SARS-CoV-2 RNA was detected in stool samples in around half of the patients [9] with COVID-19. Unfortunately, viral stool sampling was performed only for Norovirus and Adenovirus, and SARS-CoV-2 virus was not determined in stool samples of this patient. From data published so far, it is known that SARS-CoV-2 can cause ischemic colitis in patients with severe COVID-19 infection, due to hemodynamic instability and vasopressor therapy. That was not the case in our patient, who was young, with no comorbidities, with a mild course of COVID-19, and without hemodynamic instability. COVID-19 infection itself could cause coagulopathy in large and small blood vessels by causing endotheliitis and systemic inflammation [11,12], leading to various thrombotic vascular conditions, ischemic colitis being among them. An international multicenter study of COVID-19 patients undergoing endoscopy showed that around third of the patients had changes that resemble ischemic colopathy [13].

Differential diagnosis of acute, hemorrhagic colitis is diverse. Besides infectious agents, acute colitis can be caused by inflammatory bowel disease, ischemia, drugs, neutropenia, radiation, etc. [14].

Considering the age of the patient, the first manifestation of ulcerative colitis could be considered as a cause of her symptoms. Ulcerative colitis is an inflammatory bowel disease in which inflammation involves rectal mucosa, but it can also extend to more proximal parts of the large bowel. Most common symptoms of ulcerative colitis are diarrhea, rectal bleeding, tenesmus, and urgency. During endoscopy, changes seen can be mild, such as mucosal hyperemia or decreased vascular pattern, to more pronounced, such as erosions, ulcerations, or spontaneous bleeding. Changes spread continuously from rectum to more proximal parts of the colon. In this patient, endoscopic findings did not correspond completely to findings that can usually be seen in patients with ulcerative colitis. For example, rectal mucosa was almost normal [15].

Another inflammatory bowel disease that can be the cause of colitis and hematochezia is Crohn’s disease of the colon. Most common symptoms of Crohn’s disease are diarrhea and abdominal pain, but fever, fatigue, anemia, and other systemic symptoms can also appear. Endoscopic findings in Crohn’s colitis are aphthous ulcers or presence of deep ulcers that cause cobblestone appearance of the colonic mucosa. Rectum is usually spared. On abdominal CT scan in Crohn’s disease patients, discontinuous lesions can be seen, as well as bowel wall thickening reflecting the transmural nature of the inflammation [16].

Besides the SARS-CoV-2 virus, ischemic colitis can also be caused by thrombotic or embolic events within large mesenteric arteries, drugs, vasculitis, volvulus, strenuous physical activity, shock, etc. It usually affects the population older than 65 years of age, with cardiovascular comorbidities or diabetes mellitus. In younger patients, ischemic colitis is usually caused by various drugs (such as oral contraceptives, nasal decongestants, and cocaine), vasculitis, coagulation disorders, or heavy exercise. Colon is especially susceptible to ischemia due to its relatively low blood flow and its sensitivity to autonomic stimulation. The most common symptoms of ischemic colitis are hematochezia, diarrhea, abdominal pain, and urgency, and it is usually characterized with an abrupt onset of symptoms. Abdominal CT imaging and colonoscopy are diagnostic methods of choice. CT imaging shows bowel wall thickening, bowel dilation, and thumbprinting. Pneumatosis or portal vein gas can also be seen in severe cases. During colonoscopy, segmental distribution and rectal sparing are common. Hyperemia, with or without ulceration, and submucosal hemorrhage are usually found. Biopsies should be obtained, and typical findings are mucosal and submucosal oedema and hemorrhage, with erythrocyte extravasation in lamina propria [14,17].

Colitis caused by drugs can be manifested as a drug reaction with eosinophilia and systemic symptoms (DRESS) syndrome, which is characterized not only by fever, eosinophilia, skin rash, and lymphadenopathy, but also by visceral organ involvement, such as colitis. It can be caused by various drugs, but most commonly by karbamazepine. The main symptoms of DRESS colitis are diarrhea (with or without hematochezia) and abdominal pain, and symptoms usually appear weeks after exposure to the culprit drug. The course of DRESS syndrome is usually unpredictable, and delayed autoimmune sequelae can develop. During colonoscopy, ulcerations can sometimes be found. On histology, eosinophil and plasma cell infiltration is typically seen [18].

In this case, the most probable cause of the patient’s symptoms was SARS-CoV-2 infection. SARS-CoV-2 PCR tests, both in the nasopharyngeal swab and sigmoid biopsy sample, came back positive. The patient had fever and other respiratory symptoms of COVID-19, although they were mild and short term. Moreover, elevated liver transaminase levels and D-dimers also pointed to the viral cause of her symptoms. Other possible infective causes were excluded. Clinical symptoms, with the abrupt appearance of hematochezia and abdominal pain, and no prior history of gastrointestinal symptoms did not point in favor of inflammatory bowel disease, and neither did endoscopy findings. Additionally, the patient recovered completely and did not have any more disease flares, even though she was not taking any medication to maintain the disease remission. Ischemia due to thromboembolic incidents of large mesenteric arteries or veins was excluded by CT angiography and was not probable, as the patient was young and without comorbidities. Other potential causes of ischemic colitis, besides SARS-CoV-2 infection, were also excluded. Lastly, colitis caused by drugs, such as the one that appears as a part of DRESS syndrome, was not likely, as the patient denied taking any drugs prior to beginning of her symptoms. Therefore, in our patient, hemorrhagic colitis was caused either by direct cytopathic effect of viral infection in colonic mucosa or by focal ischemic changes due to coagulopathy in small blood vessels of the colon. Our finding that viral load from nasopharyngeal swab was higher than from colonic biopsy samples, was consistent with previous reports that viral loads from respiratory tract samples are usually much higher than in non-respiratory samples. High viral loads from samples from upper respiratory tract are not related with clinical manifestations and can also be found in asymptomatic patients. This can be explained by the fact that higher viral load poses a greater risk for further transmission of the virus [19]. The lower viral load in bowel mucosa in this patient shows that probably both the cytopathic effect of viral replication in epithelial cells of bowel mucosa and focal ischemia due to coagulopathy were the cause of hemorrhagic colitis.

As no conclusive data on the treatment of hemorrhagic colitis which accompanies COVID-19 was published at the time of this case, therapy was given to address potential causes and consequences of inflammatory changes in bowel mucosa. As it was suspected that both viral infection of colonic mucosa and coagulopathy of colonic small blood vessels were cause of symptoms in our patient, she was treated with corticosteroids to lower the inflammatory burden of COVID-19 infection systemically. Mesalamine is a medication used in ulcerative colitis as a systemic and topical treatment. It is shown that it reduces inflammation of the intestinal mucosa acting locally. It is a safe and well-tolerated medication, with proven efficacy in mild to moderate ulcerative colitis [20]. In this patient, we used mesalamine to lower the inflammatory burden locally in the intestinal mucosa. Small-dose low-molecular-weight heparin (LMWH) was given to address the possible coagulopathy. Metronidazole was firstly used ex juvantibus, to address the possible *C. difficile* infection. When the result came back negative, it was continued to lower the possibility of bacterial translocation from bowel lumen to bloodstream.

## 4. Conclusions

Data about COVID-19 colitis are scarce. We wanted to show that severe gastrointestinal symptoms, such as hemorrhagic colitis, can be the main presentation of COVID-19, even in young patients with no prior comorbidities. In such a case, a PCR test in biopsy samples can be performed to prove SARS-CoV-2 infection of bowel mucosa. Combination of corticosteroid therapy, mesalamine and LMWH showed to be effective in our patient.

## Figures and Tables

**Figure 1 medicina-59-00576-f001:**
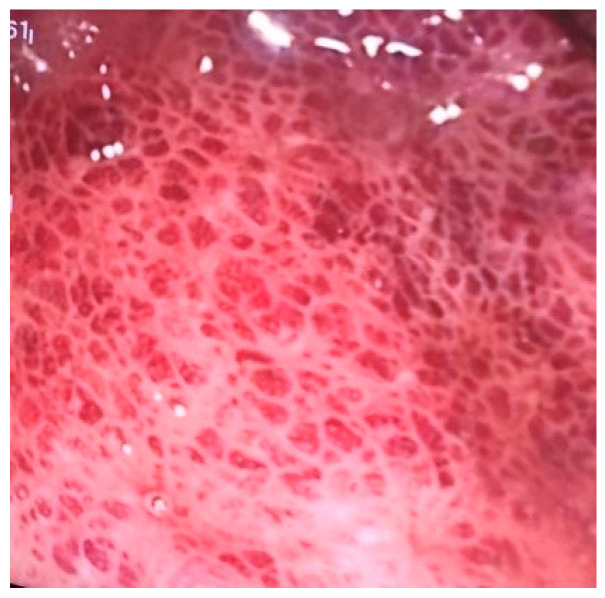
Sigmoidoscopy image of COVID-19 colitis showing submucosal bleeding.

**Figure 2 medicina-59-00576-f002:**
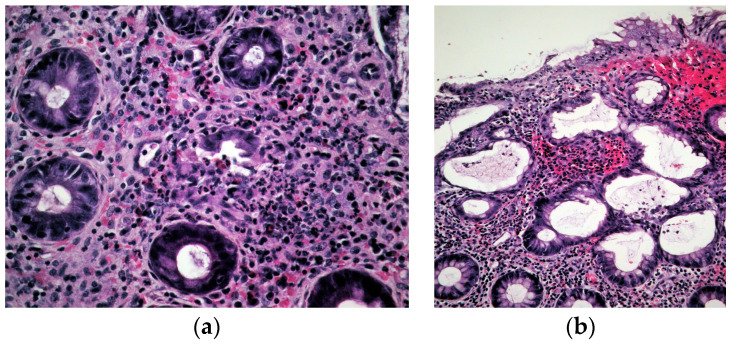
Histology of colonic mucosa: (**a**) Destructed crypt with granulocyte, eosinophil, and plasma cell infiltrate. Hematoxylin and eosin, ×400; (**b**) Neutrophil infiltrate in upper half of lamina propria in colonic mucosa. Hematoxylin and eosin, ×200.

**Figure 3 medicina-59-00576-f003:**
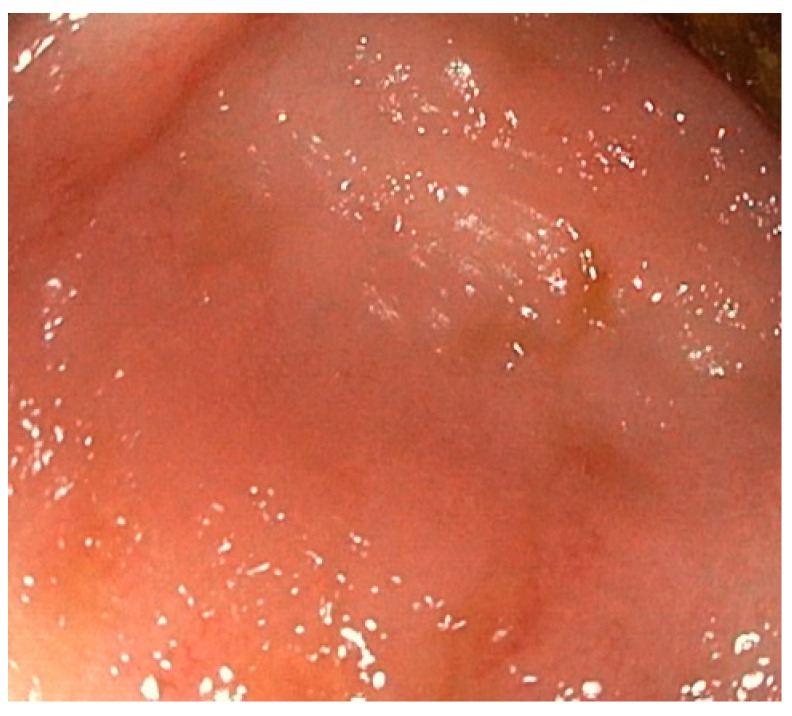
Sigmoidoscopy image of COVID-19 colitis after recovery.

## Data Availability

All data regarding the findings are available within the manuscript.

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
