# Peer review of "Colitis as the Main Presentation of COVID-19: A Case Report"

_medicina, 2023, doi:10.3390/medicina59030576_

Round 1
Reviewer 1 Report
1. Clostridioides dificile should be in italics throughout the text;
2. Why was azithromycin administered when there were no respiratory symptoms and once they appeared it was clear that it was viral infection?
3.What was the reason to administer mesalamine and metronidazole?
4. Bloody diarrhea should be changed to hematochezia
5. Salmonella, Shigella etc- please change to italics
6. Discussion should include non infectious differential diagnosis of hematochezia. For example, DRESS syndrome can present as hemorrhagic colitis and it is yet another unusual manifestation that can present the same. While in your patient DRESS syndrome is not a case, it is interesting to mention from educational perspective ( https://pubmed.ncbi.nlm.nih.gov/34575398/)
Reviewer 2 Report
Peer review regarding "Colitis as the main presentation of COVID-19: a case report"
I have read this article with great interest. The authors presented an interesting case report of a COVID-19 patient who presented with hematochezia. The purpose of the study was presented in the abstract and the title of the article corresponded to its content.
While the authors have made good work for this research, a few concerns need to be discussed or clarified.
1. The SARS-CoV-2 viral load from the rectal biopsy was lower than the respiratory specimen, while the colitis was severe and the respiratory symptoms were mild. Please discuss how the authors concluded that SARS-CoV-2 was a causative organism of colitis presented in this patient rather than an incidental finding despite contradictory findings between symptoms and viral load.
2. The authors mentioned in the discussion that SARS-CoV-2 RNA was detected in stool samples in around half of the patients, but the viral stool examination was negative for the presenting patient. Please discuss more on this topic.
3. Did the patient also get COVID-19 testing on the first visit?
Round 2
Reviewer 1 Report
I would like to thank the authors on detailed revisions. In my opinion the paper has been improved and I would recommend acceptance
Reviewer 2 Report
Thank you the authors for addressing the concerns with clear and detailed explanation.